# Identification of the Effects of 5-Azacytidine on Porcine Circovirus Type 2 Replication in Porcine Kidney Cells

**DOI:** 10.3390/vetsci11030135

**Published:** 2024-03-20

**Authors:** Yiyi Shan, Qi Xiao, Kongwang He, Shenglong Wu, Wenbin Bao, Zhengchang Wu

**Affiliations:** 1Key Laboratory for Animal Genetics, Breeding, Reproduction and Molecular Design of Jiangsu Province, College of Animal Science and Technology, Yangzhou University, Yangzhou 225009, China; mz120211434@stu.yzu.edu.cn (Y.S.); slwu@yzu.edu.cn (S.W.); wbbao@yzu.edu.cn (W.B.); 2Institute of Veterinary Medicine, Jiangsu Academy of Agricultural Sciences, Nanjing 210014, China; xiaoqi_2122@163.com (Q.X.); kwh2003@263.net (K.H.); 3Key Laboratory of Veterinary Biological Engineering and Technology, Ministry of Agriculture, Nanjing 210014, China; 4Jiangsu Co-Innovation Center for Prevention and Control of Important Animal Infectious Diseases and Zoonoses, Yangzhou University, Yangzhou 225009, China; 5GuoTai (Taizhou) Center of Technology Innovation for Veterinary Biologicals, Taizhou 225300, China; 6International Research Laboratory of Prevention and Control of Important Animal Infectious Diseases and Zoonotic Diseases of Jiangsu Higher Education Institutions, Yangzhou University, Yangzhou 225009, China

**Keywords:** 5-Azacytidine, porcine circovirus type 2, porcine kidney cells

## Abstract

**Simple Summary:**

Porcine circovirus type 2 (PCV2) infection can cause immunosuppression in pigs, create conditions for other virus infections, and bring serious economic losses to the pig industry. 5-Azacytidine (5-Aza) is a potent inhibitor of DNA methylation. In this study, we focused on the effects of 5-Aza on PCV2 infection in porcine kidney cells (PK15 cells). We found that 5-Aza could activate cell apoptosis and inflammatory factors and promote PCV2 replication through the MAPK signaling pathway.

**Abstract:**

Porcine circovirus type 2 (PCV2) is the main pathogen causing post-weaning multisystemic wasting syndrome (PMWS), which mainly targets the body’s immune system and poses a serious threat to the global pig industry. 5-Azacytidine is a potent inhibitor of DNA methylation, which can participate in many important physiological and pathological processes, including virus-related processes, by inhibiting gene expression. However, the impact of 5-Aza on PCV2 replication in cells is not yet clear. We explored the impact of 5-Aza on PCV2 infection utilizing PK15 cells as a cellular model. Our objective was to gain insights that could potentially offer novel therapeutic strategies for PCV2. Our results showed that 5-Aza significantly enhanced the infectivity of PCV2 in PK15 cells. Transcriptome analysis revealed that PCV2 infection activated various immune-related signaling pathways. 5-Aza may activate the MAPK signaling pathway to exacerbate PCV2 infection and upregulate the expression of inflammatory and apoptotic factors.

## 1. Introduction

Porcine circovirus (PCV) is a single-stranded circular non-enveloped DNA virus, belonging to the circoviridae family, and *Circovirus genus* [1,2]. PCV can be divided into four serotypes: PCV1, PCV2, PCV3, and PCV4 [3]. Among these, it has been found that PCV2 has 11 open-reading frames, among which ORF2 encodes the Cap protein in the negative strand of the genome. The Cap protein functions as the structural component and primary immunogen of PCV2, and plays a pivotal role in the replication and infection of PCV2 [4]. The interaction between the PCV2 capsid and host cell factors may trigger a conformational alteration or disassembly of the PCV2 capsid, a process essential for the translocation of the viral genome to the replication site [5]. PCV2 is known to cause postweaning multisystemic wasting syndrome (PMWS) [6,7,8,9], characterized by affected piglets aged between 8 and 16 weeks exhibiting symptoms such as jaundice, weight loss, respiratory distress, lymph node enlargement, and diarrhea [10]. It is generally believed that PCV2 infection affects the host immune system by causing lymphocyte depletion, dysregulation of cytokine secretion, immunosuppression, and increased susceptibility to secondary viral infections, thereby facilitating co-infections with other pathogens [11]. PCV2 has been clinically observed to interact with porcine reproduction and respiratory syndrome virus (PRRSV), porcine parvovirus (PPV), *Glaesserella parasuis*, or *mycoplasma*. These diseases caused by PCV2 are collectively referred to as circovirus disease (PCVAD) [12,13]. Therefore, the economic losses imposed by PCV2 on the global pig industry are considerable [14,15]. Currently, there is a lack of specific drugs targeting PCV2, with vaccination being the primary strategy for preventing PCV2 infections and transmissions. However, there are an increasing number of subtypes due to continuous variations in the PCV2 genome, and this weakens the stimulating effect of existing vaccines on the immune system and reduces their effects [16]. Therefore, we need to find drugs that can treat PCV2 to effectively control PCVAD.

Epigenetic mechanisms refer to pathways that produce reversible or heritable gene expression without altering the DNA sequence and can be affected by environmental stimuli [17]. Among these, DNA methylation stands out as a prominent epigenetic modification, whereby methylation of cytosine residues at the 5′ position yields 5-methylcytosine (5-mC) through the enzymatic action of the DNA methyltransferase (DNMT) family [18]. Viral genomic DNA methylation is involved in the regulation of gene expression and plays a pivotal role in the interaction between a virus and a host. In the Epstein–Barr virus (EBV) [19], hepatitis B virus (HBV) [20], human papillomavirus (HPV) [21], herpes simplex virus 1 (HSV-1) [22], and adenovirus type 12 (AD12) [23], DNA methylation serves as a strategy to evade host immune surveillance, impeding viral gene transcription and replication. Studies have shown that the genome of PCV2 is single-stranded circular DNA, and its complementary strand is usually encoded in host cells [24]. As a classical DNA methylation inhibitor, 5-Aza can inhibit DNA methylation by directly binding to DNA methyltransferase in the receptor genome [25,26]. Therefore, we explored the role of 5-Aza on PCV2 replication.

In this study, porcine kidney cells were used as a model to study the impacts of 5-Aza in PCV2 infection. We first determined the optimal concentration of 5-Aza without cytotoxicity and the optimal time point for virus proliferation. Secondly, we examined the effects of 5-Aza on PCV2 proliferation at 48 h. The results show that 5-Aza promotes the proliferation of PCV2. We further found that 5-Aza could promote the expression of inflammatory and apoptotic cytokines after PCV2 infection of PK15 cells. Finally, the signaling pathway of 5-Aza promoting PCV2 proliferation was screened using transcriptome sequencing, and our results suggested that 5-Aza may cause the above changes through the MAPK signaling pathway. Our study provides new information that should help to prevent the inappropriate use of 5-Aza in PCV2 infection in the future and trigger reflection on the use of 5-Aza in viral infections in general.

## 2. Materials and Methods

### 2.1. Cell Culture and PCV2 Infection

PK15 cells (ATCC, CCL-33), in a 10% fetal bovine serum (FBS) (Ozfan, Nanjing, China) and 1% penicillin streptomycin mixture (100 μg/mL penicillin, 0.1 mg/mL streptomycin) (Solarbio, Beijing, China) in DMEM (BasalMedia, Shanghai, China), were cultured in a 5% CO_2_ incubator at 37 °C. PCV2d was preserved in our laboratory.

### 2.2. Cell Activity Detection

5-Aza (5-Azacytidine, ≥98% (HPLC)) (Aladdin, Shanghai, China) was dissolved in pure water. A total of 2000 cells per well (100 μL) were seeded uniformly in the medium mixed with 10% FBS in a 96-well plate and cultured at 37 °C in a 5% CO_2_ incubator for 24 h. The complete DMEM containing the final concentration of 5-Aza was replaced with 5 μM, 10 μM, 20 μM, 30 μM, and 40 μM, and treatment time was set at 48 h. The Cell Counting Kit-8 (Vazyme, Nanjing, China) was used to detect cell viability. The CCK-8 solution was added to the 96-well plate with 10 μL per well. Optical density values were detected at 450 nm by using a Tecan Infinit200 microplate reader (Sunrise, Tecan, Switzerland).

### 2.3. Cell Immunofluorescence Staining

Cells were infected with PCV2 and then collected at different time points for experiments. Cells were washed with PBS and kept for 30 min at 37 °C after adding 4% paraformaldehyde. The cells were treated with 1% Triton X-100 for 10 min to break the membranes, and then bovine serum albumin (BSA) (Solarbio, Beijing, China) was added and blocked for 2 h at 37 °C. The BSA was discarded, and an antibody against the capsid protein Cap of PCV2 was used as the primary antibody (PCV2 Cap antibody, VMRD, Pullman, WA, USA); then, the cells were incubated in a refrigerator at 4 °C overnight. Subsequently, the cells were gently washed with PBST 3 times and then a fluorescent-labeled secondary antibody (Solarbio, Beijing, China) was added and incubated away from light for 1 h. The concentration of the primary antibody was 2.66 μg/mL, and the concentration of the secondary antibody was 125 μg/mL. After the cells were gently cleaned in PBST 3 times, DAPI was added to the cells. The different treated cells were then observed under a fluorescence microscope (Leica Microsystems, Wetzlar, Germany).

### 2.4. DNA Extraction

Whilst the cells were being infected with PCV2, mock groups were set. DNA was extracted using a Vazyme DNA extraction kit according to the instructions. Concentration and purity were determined with an ND-1000 nucleic acid/protein concentration tester, and then the cells were stored at −20 °C.

### 2.5. RNA Extraction and cDNA Synthesis

After collecting the cells, total RNA was isolated by using Trizol reagent (Takara, Shiga, Japan); the quality of the RNA was evaluated via 1% formaldehyde denatured agarose gel electrophoresis, and the RNA concentration was measured with an ND-1000 nucleic acid/protein concentration analyzer. The samples were saved in a refrigerator at −80 °C.

Reverse transcription experiments were carried out using a reverse transcription kit (Vazyme, Nanjing, China) to synthesize cDNA using the extracted total RNA as the template: the 20 μL reaction system contained 5 × qRT SuperMix II at 4 μL, 4× gDNA wiper Mix at 4 μL, 1000 ng total RNA, and RNase-free ddH_2_O supplemented to 20 μL. The reaction procedure was as follows: 50 °C for 15 min, 85 °C for 5 s, and 4 °C for preservation.

### 2.6. qPCR Detection

Based on the gene sequences published in the GenBank database, primers for qPCR were designed using Primer Premier 6.0 software, with *GAPDH* as the internal reference gene. All primers were composed using TsingKe (Beijing, China). Detailed information about each primer is shown in Appendix A. A real-time fluorescence quantification kit (Vazyme, Nanjing, China) was used for the qPCR analysis. All qPCR reactions were performed on a volume of 20 µL: 2 × SYBR Premix ExTapTM II 10 µL, PCR forward primer (10 µmol/L) 0.4 µL, PCR reverse primer (10 µmol/L) 0.4 µL, 50 × ROX reference dye II 0.4 µL, cDNA 2.0 µL, and RNase-free ddH_2_O were used to make up the total volume of 20 µL. Three independent replications were set up for each analysis sample.

### 2.7. Total Protein Extraction from Cell Samples

After infection with PCV2, cells were harvested and washed with PBS, which was pre-cooled at 4 °C, and a mixture containing a protein-cracking solution (RIPA) (Applygen, Beijing, China) and a protease inhibitor (TargetMOI, MA, USA) was added. After the cells were cracked at low temperatures for 20 min, the supernatant was obtained after centrifugation at 14,000 rpm at 4 °C for 20 min. The protein concentration was adjusted using the BCA Protein Assay Kit (Beyotime, Shanghai, China) and denatured at 98 °C for 10 min with a 5 × loading buffer. The denatured samples were then stored at −20 °C.

### 2.8. Western Blotting

The treated samples were separated using 10% sodium dodecyl sulfate–polyacrylamide gel and electrotransferred to a polyvinylidene fluoride (PVDF) membrane (Millipore, Canada, USA). After blocking for 2 h at room temperature, the primary antibody (PCV2 Cap antibody, GeneTex, TX, USA; HSP90 antibody, Proteintech, Wuhan, China) was incubated at 4 °C overnight. The PVDF membranes were rinsed using TBST, which was incubated with a secondary antibody (Proteintech, Wuhan, China) at room temperature for 2 h. The concentration of the primary antibody was 1 μg/mL, and the concentration of the secondary antibody was 0.02 μg/mL. After washing with TBST, the samples were developed and analyzed via ECL.

### 2.9. Reactive Oxygen Species Assessment

Reactive oxygen species (ROS) levels were determined with a 2′7′-dichlorofluorescein diacetate (DCFH-DA) probe using a reactive oxygen species assay kit (Solarbio, Beijing, China). The cells were removed from the incubator and serum-free DMEM containing a diacetate (DCFH-DA) probe was added, and they were incubated for 30 min. The fluorescence emitted by the cells was observed under a fluorescence inverted microscope, and changes in the ROS in the cells before and after adding 5-Aza were detected.

### 2.10. Flow Cytometry Analysis

The Annexin V-FITC/PI apoptosis detection kit (Solarbio, Beijing, China) was used to detect the changes in apoptosis levels of the PK15 cells and PK15 + 5-Aza-treated cells. The treated cells were digested into a centrifuge tube with trypsin without EDTA; the supernatant was removed through centrifugation. The single-staining tube group was used as a positive control, and 5 µL Annexin V/FITC was added and mixed, before incubation at room temperature in a dark place for 5 min. The solution without the Annexin V/FITC and propidium iodide (PI) solution was used as the negative control group. Annexin V/FITC and PI were added to the treatment group. Cell apoptosis was measured using a FACScan flow cytometer (Becton Dickinson, CA, USA). CytExpert 2.3 and FlowJo 7.6 software were used for analysis.

### 2.11. RNA Sequencing

After 48 h of PCV2 treatment, PK15 cells in the control group (NC, n = 3), PCV2 group (PCV2, n = 3), and PCV2 + 5-Aza group (PCV2 + 5-Aza, n = 3) were collected. Total RNA was extracted using Trizol reagent (Takara, Kusatsu, Japan), and the quality of the RNA was assessed via 1% formaldehyde denaturing agarose gel electrophoresis. RNA concentration was determined using an ND-1000 nucleic acid/protein concentration analyzer. Subsequently, all RNA samples were converted into double-stranded cDNA and subjected to sequencing on the Illumina Hiseq2500 platform from the Oebiotech corporation (Shanghai, China). A corrected *p*-value < 0.05 and |log_2_(foldchange) < 1| were set as the identification principles of differentially expressed genes (DEGs). GOseq was used to annotate the Gene Ontology (GO) function of DEGs on the returned data. The KOBAS software (version 3.0) was used to detect the enrichment of DEGs in KEGG (Kyoto Encyclopedia of Genes and Genomes) pathways. GO terms and KEGG pathway analyses were performed using corrections with a cutoff value of 0.05.

### 2.12. Data Processing and Analyses

The 2^−ΔΔCt^ method was used to calculate the relative quantitative data, and the expression level was homogenized with internal reference genes. SPSS 25.0 software (SPSS, Inc., Chicago, IL, USA) was used for statistical analyses. The *p*-values were calculated using a two-group unpaired *t*-test or a one-way analysis of variance. All data were expressed as the mean ± SD of three repeated samples. After statistical treatment, *p* < 0.05 indicated that the difference was statistically significant.

## 3. Results

### 3.1. Working Concentration of 5-Aza and Drug Safety Evaluation

To explore the optimal concentration of 5-Aza that does not impair cell activity, we compared the cell activity of PK15 cells treated with various concentrations of 5-Aza (0, 5, 10, 20, 30, and 40 μM) for 48 h in vitro. We found that high concentrations of 5-Aza reduced PK15 cells activity and induced cell injury. When the concentration of 5-Aza was below 10 μM, the cell activity was not significantly decreased in the treatment group, but when the concentration of 5-Aza was higher than 20 μM, the cell activity was significantly decreased (Figure 1A). The change in intracellular basic ROS levels is an important indicator of healthy mitochondrial function. We detected changes in ROS levels in the PK15 cells treated with 5-Aza. Upon comparing them with those in the negative control group, it was found that there was no significant change in the ROS fluorescence signal in the 10 μM 5-Aza-treated group (Figure 1B,C). The detection of cellular ROS with DCFH-DA probes showed the same results (Figure 1D). Next, we detected Occludin, which plays an essential role in regulating cell morphology, division, and signaling, and is a necessary part of the cytoskeleton system [27]. The indirect immunofluorescence assay (IFA) indicated that 10 μM 5-Aza did not cause a significant reduction in Occludin at 48 h (Figure 1E). There were no apparent changes in the mRNA levels of various pro-inflammatory factors *(IFN-α*, *IL-16*, *IL-18*, *IL-6*, and *TNF-α*) after treatment with 10 μM 5-Aza (Figure 1F). In summary, the treatment of PK15 cells with 5-Aza at a concentration of 10 μM had no toxic effect on the cells, so this condition was selected for subsequent experiments.

### 3.2. Establishment of a PCV2-Infected PK15 Cell Model

After the cells’ infection with PCV2, Cap expression was measured via qPCR analysis, Western blot analysis, and IFA at 0, 12, 24, 36, 48, and 72 h, respectively. As shown in Figure 2, the expression and distribution of PCV2 Cap gradually increased with the extension of time points after PCV2 infection in the PK15 cells. Cap expression increased gradually from 0 to 48 h, reaching a maximum at 48 h, while it was decreased at 72 h compared with the level at 48 h (Figure 2A,B). In the IFA, the staining of the viral protein Cap also showed an increase in fluorescence (shown in green in Figure 2C), reaching a maximum after 48 h.

### 3.3. Effects of 5-Aza on the Replication Level of PCV2

The effects of 5-Aza on the PCV2 infection of PK15 cells was investigated using qPCR, IFA, and Western blot tests. As shown in Figure 3A, this experiment was set up for a total of four groups, including the negative control group, the 5-Aza treatment group, the PCV2 treatment group, and the PCV2 + 5-Aza treatment group, with three replicates in each group. The results showed that 10 μM 5-Aza significantly enhanced PCV2 *Cap* gene expression (Figure 3A). The Western blot results indicated that the replication level of PCV2 was significantly upregulated under the treatment of 10 μM 5-Aza (Figure 3B). The IFA results further confirmed the presence of a large amount of red fluorescence of the PCV2 capsid protein in the PCV2 + 5-Aza-treated group (Figure 3C). Thus, the amount of PCV2 Cap protein increased after 5-Aza treatment. In conclusion, 5-Aza was able to promote PCV2 proliferation in PK15 cells at 48 h.

### 3.4. Effects of 5-Aza on Apoptosis and Cytokine Levels in PCV2 Infection

The qPCR results showed that cellular pro-inflammatory factors are also involved in the biological process of PCV2 infection in cells. PCV2 infection of PK15 cells stimulated the expression of *IL-6*, *IL-1β*, *IL-12*, *IFN-α*, and *IFN-β*, while 5-Aza treatment aggravated the expression of *IL-1β*, *IL-12*, *IFN-α*, and *IFN-β*, but with no significant changes in *IL-6* (Figure 4A).

Additionally, PCV2 infection obviously induced the PK15 cells’ apoptosis. After 5-Aza treatment, the mRNA levels of the apoptosis-related genes *BAX*, *Caspase3*, and *Caspase8* were further upregulated compared with those in the PCV2 treatment group, while *BCL-2* and *Caspase9* exhibited no significant changes (Figure 4B). The results of the Western blot testing showed that the PCV2 + 5-Aza treatment group exhibited further increased secretion of apoptosis-related proteins BAX, Caspase8, and Caspase3, while the expression of BCL-2 was downregulated (Figure 4C).

In conclusion, the addition of 5-Aza to the PCV2-infected PK15 cells intensified apoptosis and inflammation.

### 3.5. Transcriptome Sequencing Analysis before and after 5-Aza Treatment and PCV2 Infection

#### 3.5.1. RNA Sequencing of Relevant Genes and Signal Pathways following PCV2 Infection

To identify the underlying mechanisms resulting from 5-Aza treatment, we performed RNA sequencing. After PCV2 infection of the PK15 cells, differential gene expression analysis showed that 2397 genes were upregulated and 1728 genes were downregulated (Figure 5A and Appendix A). The differential gene-grouping cluster map showed a striking difference between the mock group and the PCV2-treated group (Figure 5B). The KEGG enrichment analysis indicated that PCV2 mainly regulated ten signaling pathways, such as the viral protein interactions with cytokines and cytokine receptor pathways and the TNF signaling pathway (Figure 5C and Appendix A). The GO enrichment analysis was performed on the differentially expressed genes, and the functions were described. After the differentially expressed genes were obtained, the differential gene attributes could be divided into three broad categories: biological processes, cellular components, and molecular functions (Figure 5D).

#### 3.5.2. Comparative RNA Sequencing Analysis of PK15 Cells Infected with PCV2 and Cells Treated with PCV2 + 5-Aza

The differentially expressed gene statistics histogram showed that the expression of 3340 genes was increased and the expression of 3314 genes was reduced between the PCV2 + 5-Aza treatment group and the PCV2-treated group (Figure 6A and Appendix A). The differential gene-grouping cluster map showed a complete separation between the two groups of samples (Figure 6B). The KEGG enrichment analysis indicated that various genes involved in the cell cycle, the FoxO signaling pathway, and cytokine–cytokine receptor interactions were enriched in the drug treatment group and the PCV2 infection group (Figure 6C and Appendix A).

#### 3.5.3. The Addition of 5-Aza Intensifies the Activation of Inflammatory Cytokines and the MAPK Signaling Pathway

Then, a joint analysis was carried out on the mock group, PCV2 treatment group, and PCV2 + 5-Aza treatment group. First, 358 genes were found to be co-increasing and 280 genes were found to be co-dropping in the three groups. In addition, 478 genes showed different trends in the two comparison groups (Figure 7A). KEGG enrichment analysis of all differential genes in the combined analysis indicated that DEGs were primarily enriched in cytokine–cytokine receptor interactions, the calcium signaling pathway, and the MAPK signaling pathway (Figure 7B and Appendix A). Previous studies have reported that the MAPK pathway is closely related to the cellular inflammatory response. As detected in the qPCR tests, *CCL1*, *CCL22*, and *MAP2K1* were significantly upregulated after 5-Aza treatment in the PCV2-infected PK15 cells (Figure 7C).

## 4. Discussion

PCV2-induced PCVAD has caused significant economic losses in global pig-breeding production. At present, vaccination remains the primary method for preventing PCV2 infection. However, due to the occurrence of strain drift and the emergence of new subtypes over time, the existing vaccines are challenged to effectively prevent and control infections as new variants emerge. Therefore, there is a critical need to explore potential therapeutic agents that can inhibit PCV2. In this study, we aimed to investigate whether 5-Aza has a therapeutic effect on PCV2-induced PCVAD. Surprisingly, our results revealed that 5-Aza could promote PCV2 replication in PCV2-infected PK15 cells. PCV2 isolates can be divided into eight genotypes (PCV2a, PCV2b, PCV2c, PCV2d, PCV2e, PCV2f, PCV2g, PCV2h, and PCV2i) based on sequence analysis [28,29]. Since 2014, the different PCV2 genotypes have been basically replaced by PCV2d, and PCV2d has become the main prevalent genotype with strong replication ability. So, we used representative PCV2d strains in our study [30,31]. However, while our study focused on PCV2d strains, it is important to acknowledge that different genotypes of PCV2 may exhibit varying levels of virulence. Future investigations will incorporate diverse PCV2 genotypes to enhance the comprehensiveness of our findings.

Although many studies on DNA viral genome methylation have been reported, different viruses have different methylation statuses, which depend on the viral life cycle stage, host and target organs, flanking nucleotide motifs, and other factors. Thus, each viral methylation also has different effects on the virus itself and the host cell [32]. PCV2 is a small DNA virus that relies on the replication mechanism of the host cell for replication. So far, there have been no reports on the genome methylation of PCV2. However, in a recent study of infectious spleen kidney necrosis virus (ISKNV, a DNA virus) in fish, we found that 5-Aza reduced the methylation level of hypermethylated CpG sites in the ISKNV genome and reduced viral DNA and protein levels in the host cells in a dose-dependent manner [33]. This highlights the potential of 5-Aza to influence DNA viral methylation patterns. Subsequent studies will delve into the impact of 5-Aza on PCV2 genomic and host cell methylation. Furthermore, PCV2 infection regulates multiple cellular processes, including cytoskeleton turnover, stress responses, macromolecular biosynthesis, energy metabolism, signal transduction, gene regulation, and immune responses [34,35,36,37,38,39,40,41]. Our examination of various time points during PCV2 infection in PK15 cells revealed peak replication levels at 48 h, coinciding with the induction of inflammation and apoptosis, which is consistent with previous reports.

Mitogen-activated protein kinases (MAPKs) are important transmitters of information from the cell surface to the nucleus. The JNK and p38 MAPK signaling pathways are essential in stress reactions such as inflammation and apoptosis [42,43]. PCV2 can activate the apoptosis signal-regulating kinase 1 ASK1 in PK15 cells [44], which regulates the MAPK JNK/SAPK and p38 MAPK signaling pathways, thus activating the phosphorylation of downstream targets c-Jun and ATF-2. Notably, inhibition of the p38 MAPK has been demonstrated to downregulate essential biological processes, including transcription and translation of the PCV2 genome within infected cells, while also diminishing the activities of apoptosis-related proteins like caspase8 and caspase3 [45,46]. In this study, the addition of 5-Aza promoted PCV2 replication and enhanced caspase8 and caspase3 activities. Transcriptome sequencing analyses have provided insights that support the notion that 5-Aza may indeed regulate MAPK signaling pathways, consequently bolstering PCV2 replication within host cells.

In addition, the intricate relationship between the MAPK signaling pathways and pro-inflammatory cytokines adds another layer of complexity to the understanding of host–virus interactions during PCV2 infection. Previous studies have highlighted the regulatory role of the p38 MAPK pathway in modulating the levels of pro-inflammatory factors following PCV2 infection. For instance, PCV2 can induce mRNA expression of IL-1β and IL-6 in PK15 cells through the ERK, JNK, and p38 MAPK signaling pathways [47], promote the phosphorylation of USP21, and inhibit the transcription of STING’s K63 ubiquitination and IFN-β, thereby inhibiting the innate immune response [48]. In this study, we detected the expression of inflammatory factors in PCV2-infected PK15 cells at 48 h. Our findings revealed that PCV2 infection resulted in the upregulation of various inflammatory mediators, including IL-6, IL-1β, IL-12, IFN-α, and IFN-β. Intriguingly, the addition of 5-Aza further augmented the expression levels of IL-1β, IL-12, IFN-α, and IFN-β, suggesting a potential role for 5-Aza in exacerbating the inflammatory response induced by PCV2 infection.

These results underscore the interplay between PCV2, host immune responses, and epigenetic modulator 5-Aza in shaping the inflammatory milieu within infected cells. Further investigations into the molecular mechanisms underlying the interaction between 5-Aza, MAPK signaling pathways, and inflammatory cytokine production will be crucial for unraveling the complex immune responses elicited by PCV2 and exploring potential therapeutic strategies to mitigate the detrimental effects of PCV2-induced inflammation in swine populations.

## 5. Conclusions

In summary, this study obtained preliminary results showing that 5-Aza significantly enhances PCV2 infection in PK15 cells. In addition, 5-Aza may increase the proliferation of PCV2 by activating MAPK signaling pathways. It regulates the downstream target proteins through the MAPK signaling pathway to affect cell inflammation, and apoptosis. These findings indicate that 5-Aza may not be suitable as an effective therapeutic agent for targeting PCV2 infection.

## Figures and Tables

**Figure 1 vetsci-11-00135-f001:**
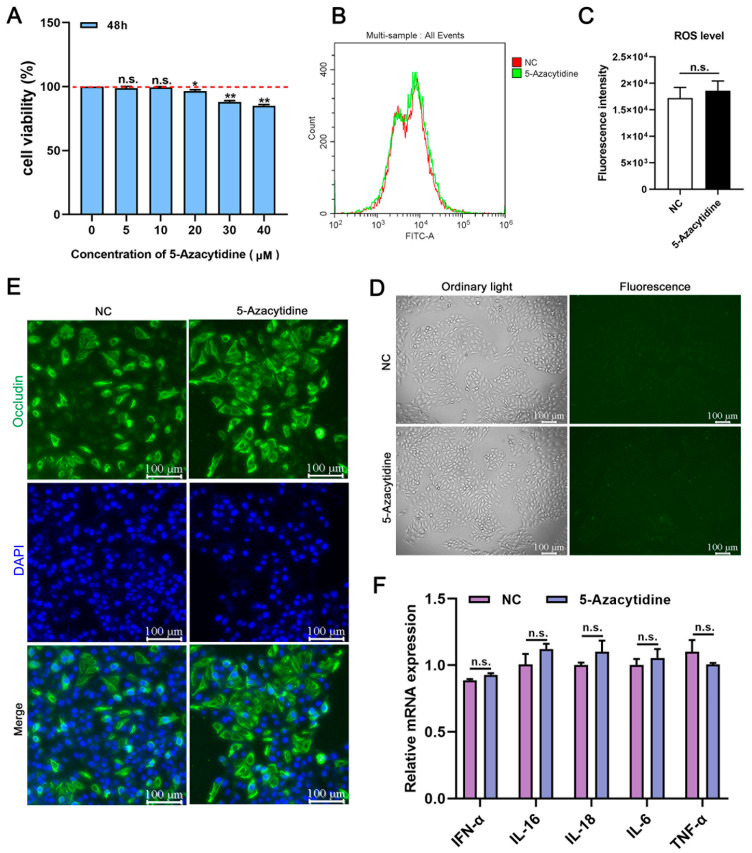
Drug safety evaluation of 5-Aza. (**A**) A CCK8 assay was used to detect the viability of cells treated with different concentrations of 5-Aza. (**B**,**C**). The red dashed line represents the standard line. Flow cytometry was used to detect the level of ROS in cells treated with 10 μM 5-Aza and the negative control group (PK15 cells). (**D**) Typical fluorescence images of ROS in PK15 cells induced by 10 μM 5-Aza. Scale bar: 100 μm. (**E**) IFA was used to detect the changes in Occludin in PK15 cells treated with 10 μM 5-Aza and the negative control group. Green fluorescence represents the protein fiber lattice structure present in the cytoplasm bound by Occludin antibodies, and blue fluorescence represents the DAPI-stained nucleus. Scale bar: 100 μm. (**F**) qPCR was used to detect the effects of 10 μM 5-Aza in the treatment group and negative control group on pro-inflammatory factors in PK15 cells. The *p*-values were calculated using a one-way analysis of variance. Data are expressed as the mean ± standard deviation (SD) of three repeats. *, **, and ^n.s.^ indicate, relative to the negative control, *p* < 0.05, *p* < 0.01, and *p* > 0.05.

**Figure 2 vetsci-11-00135-f002:**
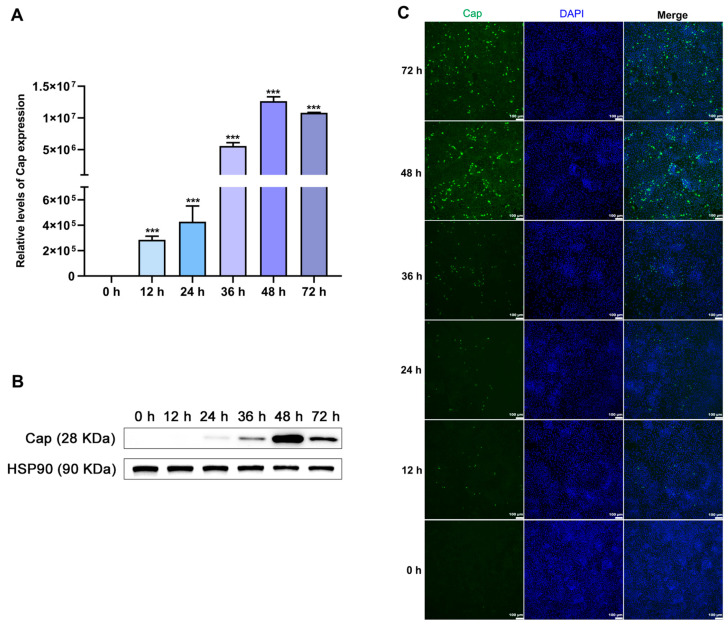
Establishment of a PCV2-infected (MOI = 1) PK15 cell model: (**A**) Relative levels of *Cap* expression when PK15 cells were infected with PCV2 at different time points. (**B**) Protein levels of Cap in PCV2-infected PK15 cells at different time points. (**C**) The number of positive cells infected with PCV2 was detected with immunofluorescence. Green fluorescence represents the Cap protein and blue fluorescence represents the DAPI-stained nucleus. Scale: 100 μm. All data are presented as the mean ± standard deviation of three repeats. *** denotes, relative to 0 h, *p* < 0.001.

**Figure 3 vetsci-11-00135-f003:**
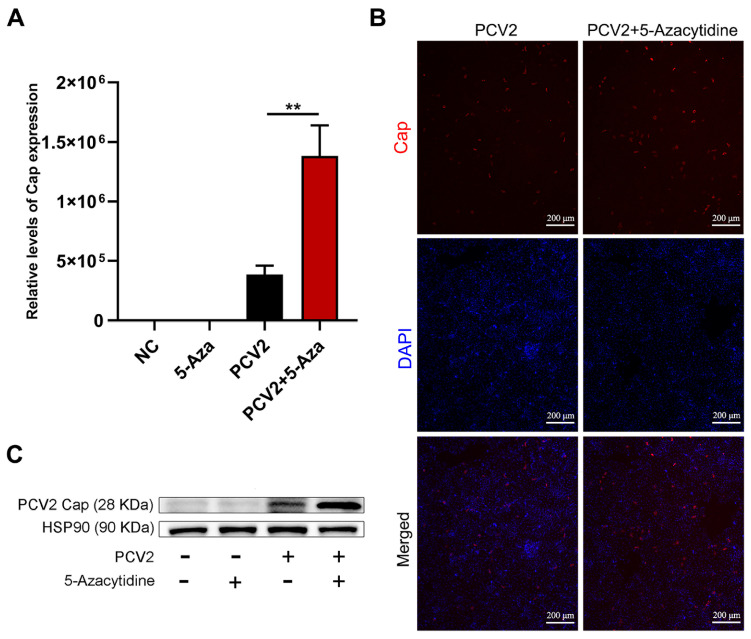
Impact of 5-Aza on the replication of PCV2 (MOI = 0.1) in PK15 cells: (**A**) The expression levels of PCV2 *Cap* after 5-Aza treatment were detected using qPCR. (**B**) The expression levels of the viral Cap protein after 5-Aza treatment were tested using Western blot analysis. (**C**) Immunofluorescence images: the red fluorescence represents the Cap protein and the blue fluorescence represents the DAPI-stained nucleus. Scale: 200 μm. All data are presented as the mean ± standard deviation of three repeats. ** indicates, relative to the PCV2-treated group, *p* < 0.01.

**Figure 4 vetsci-11-00135-f004:**
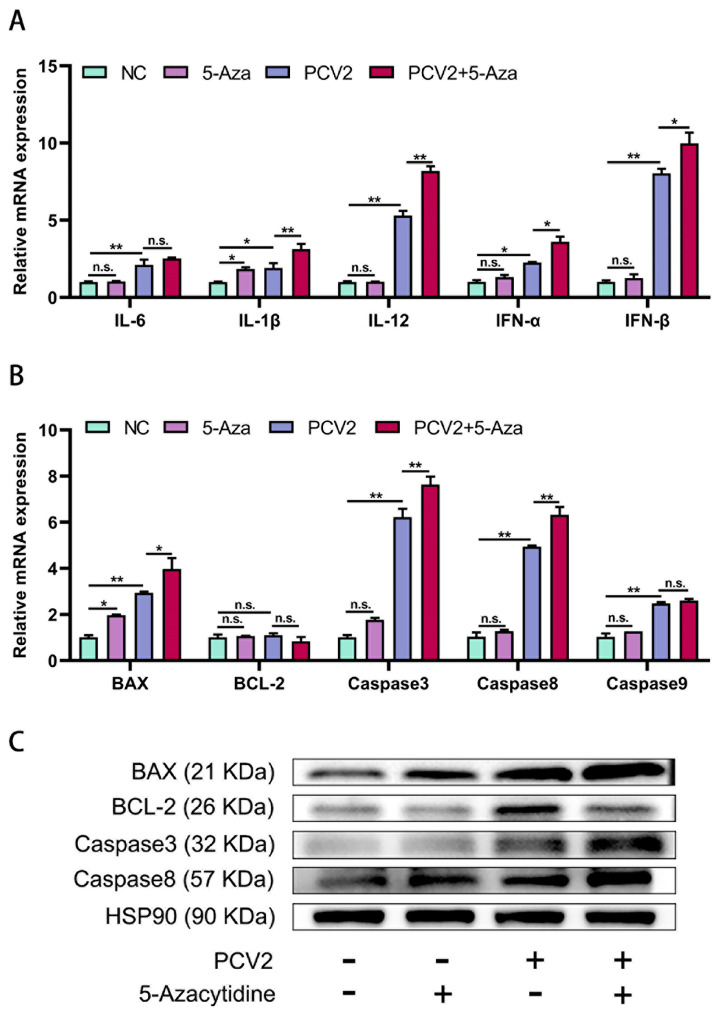
5-Aza can promote the expression of inflammatory and apoptotic factors after PCV2 infection. (**A**) qPCR was used to detect the mRNA levels of *IL-6*, *IL-1β*, *IL-12*, *IFN-α*, and *IFN-β*. (**B**) qPCR was used to detect the mRNA levels of *BAX*, *BCL-2*, *Caspase3*, *Caspase8*, and *Caspase9*. (**C**) Protein levels of apoptosis-related cytokines were detected with Western blot testing after 5-Aza treatment. All data are presented as the mean ± standard deviation of three repeats. *, **, and ^n.s^ indicate, relative to the negative control group, relative to the PCV2+ 5-Aza-treated group, or relative to the PCV2-treated group, *p* < 0.05, *p* < 0.01, and *p* > 0.05.

**Figure 5 vetsci-11-00135-f005:**
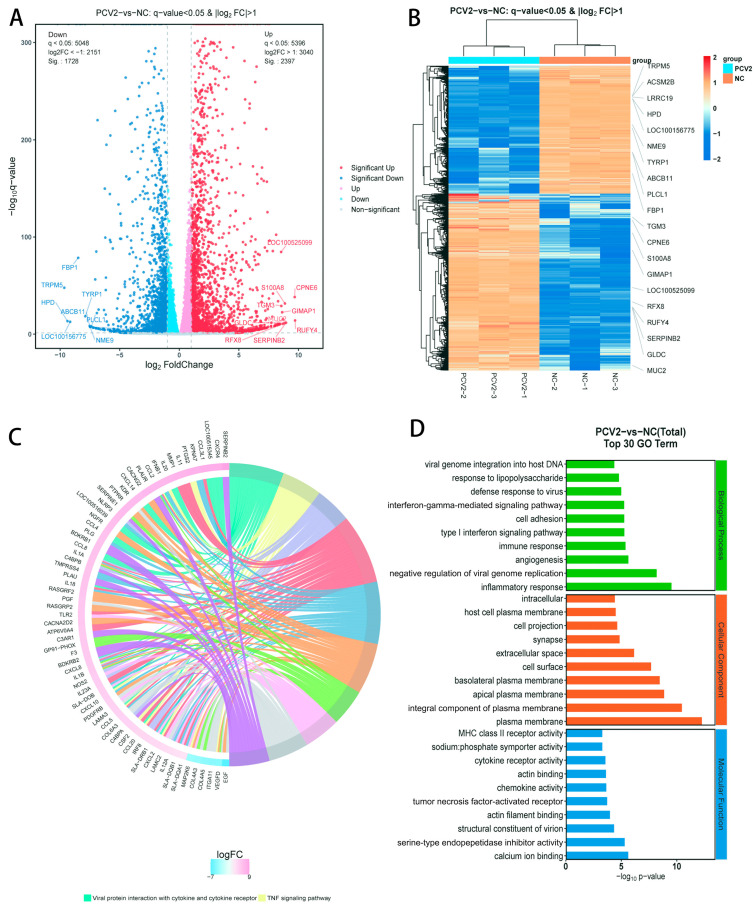
Analysis of the RNA sequencing profiles and differentially expressed genes (DEGs) of the mock group and PCV2-treated group. (**A**) Volcano plot of differentially expressed genes. Blue plots represent downregulated genes and red plots represent upregulated genes. (**B**) Differential gene-grouping cluster map, where red indicates a relatively high expression of protein-coding genes, and blue indicates a relatively low expression of protein-coding genes. (**C**) Kyoto Encyclopedia of Genes and Genomes (KEGG) analysis indicates the enrichment of 10 signaling pathways of differential genes. (**D**) Gene Ontology (GO) analysis indicates that the DEGs could be divided into three categories: biological processes, cellular components, and molecular functions.

**Figure 6 vetsci-11-00135-f006:**
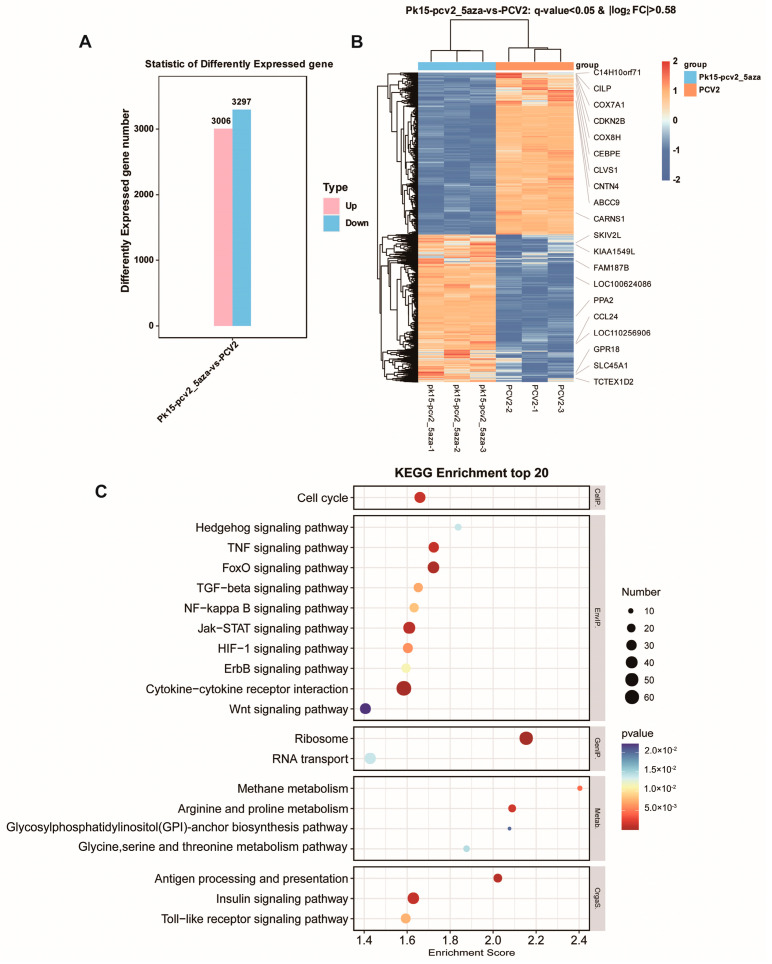
RNA sequencing profile and DEG analysis of the PCV2 treatment group and PCV2 + 5-Aza treatment group. (**A**) Statistical histogram of differentially expressed genes. On the horizontal axis are the PCV2 and PCV2 + 5-Aza groups; the vertical axis represents the number of differential genes in the comparison group. (**B**) Differential gene-grouping cluster map. Red: relatively high expression of protein-coding genes, blue: relatively low expression of protein-coding genes. (**C**) The bubbles represent the top 20 enriched KEGG pathways of the DEGs. The size of the dot represents the level of DEG enrichment. The color of the dot represents the significance of the DEG enrichment.

**Figure 7 vetsci-11-00135-f007:**
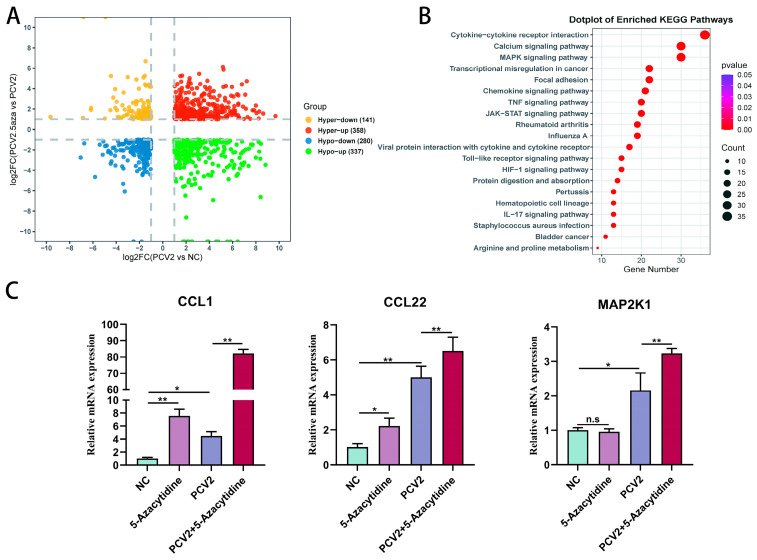
Differentially expressed genes (DEGs) in the mock group, PCV2 treatment group, and PCV2 + 5-Aza treatment group were analyzed jointly. (**A**) The four-quadrant map shows the distribution of differential genes in different groups. The gray dashed line represents the |log2FC| =1 for the difference analysis. (**B**) KEGG pathway analysis of the DEGs. (**C**) qPCR was used to detect the expression of differential genes. All data are presented as the mean ± standard deviation of three repeats. *, **, and ^n.s^ indicate, relative to the negative control group, the PCV2+ 5-Aza-treated group, or the PCV2-treated group, *p* < 0.05, *p* < 0.01, and *p* > 0.05.

## Data Availability

The raw data supporting the conclusions of this article will be made available from the corresponding author, without undue reservation.

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
