# Peer review of "Identification of the Effects of 5-Azacytidine on Porcine Circovirus Type 2 Replication in Porcine Kidney Cells"

_vetsci, 2024, doi:10.3390/vetsci11030135_

Round 1

Reviewer 1 Report (Previous Reviewer 5)

Comments and Suggestions for Authors

All concerns have been addressed.

Author Response

Thanks for your comments.

Reviewer 2 Report (Previous Reviewer 2)

Comments and Suggestions for Authors

Previous comments have been addressed.

Author Response

Thanks for your comments.

Reviewer 3 Report (New Reviewer)

Comments and Suggestions for Authors

In this paper, the authors investigate the effect of 5-Azacytidine on Porcine Virus 2 infected porcine kidney cells, concentrating on virus replication and the cellular response to the combination of infection and 5-Aza treatment and observe that 5-Aza does not impair but rather increase viral replication as well as the production of inflammatory cytokines and apoptosis markers by the cells. Thus, the results presented here are of relevance to the field since they show that the use of 5-Aza to treat PCV2 infection is rather counterproductive.

However, although the topic of the paper is important, the work itself presents some points which need to be addressed. Some controls to be added, questions to experiments and the project as well as a few incomprehensible sentences are included in the joint word file. 

Comments on the Quality of English Language

English was mostly OK in the first part but some sentences in the discussion are not understandable (see word file). For the minor points/ English mistakes listed, note that I have probably missed some. Moreover, some things are formulated in an imprecise way which makes them wrong (mRNA expression instead of level, MAPKs are a pathway, translation of PCV2...). Therefore, please let a native speaker read and check the paper.

Author Response

Reviewer 3 comments

Major points

In the introduction, you emphasize that one problem with PCV2 is the high number of subtypes. L30, you mention PCV2d but haven’t told us yet that subtypes exist and how many there are. L70, you tell us you will use PCV2d in experiments but later on the virus is named PCV2 without “d”. Did you use subtype d for all experiments? Would it make a difference if you had used another one? Perhaps that is something to mention in the discussion?

Response:

Thanks for your advice. I have revised the contents of the Discussion section and explained the questions you raised. (L356-364)

In Fig 1 B to F, a positive control is missing to verify that the experiments are actually working. A third point with one of the higher concentrations of 5-Aza for which an increase in ROS (B/C/D), a reduction in occluding (E) and increase in pro-inflammatory factors (F) are seen is needed.

Response:

Thank you for reminding us. In the preliminary experiment, we treated the cells with low to high concentrations of 5-Aza, measured cell viability, and confirmed the appropriate dosage of 5-Aza (Figure1A). Therefore, high concentration treatments were not set in subsequent experiments. However, your advice is valuable. We will refine the experimental protocol to be more rigorous in the future. 

You hypothesize in the introduction that DNA methylation of the viral DNA might change upon 5-Aza treatment. Does it change? Did you check?

Response:

Sorry for the confusing. I have supplemented relevant references in the Introduction and Discussion sections to support my ideas. (L71-75 and L365-374)

In Fig 2A, after 48h of infection you reach a viral titre of 1,1 107, while in Fig 3 you reach about 4*105 forPCV2 and 1,4 106 for PCV2 + 5-Aza which is much less than in Fig2A, even when 5-Aza is added. Can you explain where the difference comes from? Also in both cases, check the unit - shouldn’t it be genome copies instead of relative levels of Cap mRNA expression?

Response:

Thanks for your suggestion. I have added the virus titer in the notes of Figure 2 and Figure 3. In addition, I have changed the "Relative levels of Cap mRNA expression" to "Relative levels of cap expression" in Figure 2A. Because I examined the copy number of the PCV2 Cap gene in the relative quantification manner.

Wouldn’t you expect to have less cells in the IFA after 48 h of PCV2 infection + 5-Aza treatment (figure 3C) since you show an increase in apoptosis in later experiments? (or even in the cells infected with PCV2 even without 5-Aza after 48 and 72 h (figure 2C)?)

Overall figure legends are too short and don’t give enough information on the experiments performed – for example, in fig 4 please tell in the legend what NC is and whether the mRNA expression is relative to NC or an internal housekeeping gene. Also, relating to the previous point, it would be appreciable to state with how much virus how many cells were infected in fig 2 and 3 so that, if the difference in titre after 48 h comes from a different input, the reader will understand.

Response:

Sorry for the misunderstanding, I have changed Figure 3C to make the results more intuitive. As can be seen in the Figure 2C, the number of cells at 0h was significantly higher than that after PCV2 infection, especially at 72h. At the same time, I improved the information of the overall figure legends, and explained what NC (negative control group) is, and the mRNA expression is relative to NC.

L 287: “differential gene expression analysis showed that gene 2397 was upregulated and gene 1728 was downregulated” should be changed to “differential gene expression analysis showed that 2397 genes were upregulated and 1728 genes 1728 were downregulated”

Response:

Thanks for your advice. I have corrected my mistake. (L294)

There are three sentences which are not understandable:

1) L364/365: The transcriptome sequencing results are conjoint analysis, we speculated that 5-Aza by influencing the MAPKs signalling pathway, and to promote a further replication of PCV2, increased cell apoptosis.

2) L369-371: Some studies have been covered that the p38 MAPK signalling pathway can modulate the mRNA level of cell-related pro-inflammatory factors, Up-regulation of inflammatory factors proteins after PCV2 infection.

3) L398-399: In order to further verify the mechanism of 5-Aza promoting PV2 replication, RNA-seq was used to screen the MAPKs signalling pathway of PCV2 replication promoted by 5-Aza.

Response:

Thank you for reminding us. “3) L398-399: In order to further verify the mechanism of 5-Aza promoting PV2 replication, RNA-seq was used to screen the MAPKs signalling pathway of PCV2 replication promoted by 5-Aza.” The contents of this section have been removed. The remaining two problems have been solved. (L390-394 and L397-399)

The discussion contains lots of repetitions from the introduction, results and within the discussion itself. This part would benefit from being rewritten and including more discussion on some points like the subtypes, the viral DNA methylation in presence of 5-Aza, the use of 5-Aza for other viral infections or alternatives to it for PCV2 infections discussed.

Response:

Thank you for your valuable advice. I have rewritten the Discussion section.

Minor points –

L35-36: you don’t need the sentence “It’s speculated that 5-Aza may exert the above effects by activating the MAPKs signalling pathway” since you already say it on l33.

Response:

Sorry for the confusing. I've rewritten it. (L36-37)

L74: The results showed that 5-Aza could promotes proliferation

Response:

Thanks for your advice. I have corrected the problem. (L84)

L76: after PCV2 infection with of PK15 cells

Response:

Sorry for the confusing. I have corrected the problem. (L86)

L78: 5-Aza May may cause

Response:

Thank you for reminding us. I have corrected the problem. (L88)

L79-81: the sentence “Our study provides a new reference and method for the prevention and treatment of PCV2, and may provide some necessary references for the treatment of PCV2 in the future” is a little bit overdone since a new treatment is not found here. Please rephrase maybe into something like “Our study provides new information should help to prevent inappropriate use of 5-Aza in PCV2 infection in the future und triggers reflection on the use of 5-Aza in viral infection in general”.

Response:

Thanks for your advice. I have replaced the original sentence. (L89-91)

L170: replace “, and Annexin V/ FITC was not added at the same time” with “a control without Annexin V/ FITC was done in parallel”

Response:

Sorry for the confusing. I've rewritten it. (L177-178)

L200: with varient various concentrations

Response:

Thank you for reminding us. I have corrected my mistake. (L205)

L202: cell injure injury 

Response:

Thank you for your valuable advice. I have corrected the problem. (L207)

L203: not obviously significantly decreased

Response:

Thanks for your advice. I have corrected my mistake. (L208)

L209: Similarly, detection of cellular ROS…

Response:

Thank you for your valuable advice. I have corrected the problem. (L214)

L212: IFA – when used for the first time write immunofluorescence assay/analysis in full

Response:

Thanks for your advice. I've given the full name. (L217)

L229: when used for the first time say that Cap is the viral capsid protein

Response:

Thank you for reminding us. I have explained in the Introduction section that Cap is the capsid protein of PCV2. (L43-46)

L233: can you explain a bit more, replacing “The green fluorescence signal represents the PCV2 Cap protein” by “in IFA, the staining of the viral protein Cap also shows an increase in fluorescence (in green in Figure 2C) to reach a maximum after 48 hours”

Response:

Sorry for the confusing. I've rewritten it. (L240-242)

L263: showed that Ccellular pro-inflammatory 

Response:

Thanks for your suggestion. I have corrected the problem. (L269)

L265: the secretion expression of IL-6

Response:

Thank you for your correction. I have corrected my mistake. (L271)

L269: caspaes3 and caspaes8 -> caspase 3 and caspase 8

Response:

Sorry for the confusing. I've rewritten it. (L275)

L286: infection in of PK15 cells

Response:

Thank you for reminding us. I have corrected the problem. (L294)

L288: difference of between the mock group from and the PCV2 treated group

Response:

Thanks for your advice. I've rewritten it. (L296-297)

L291: the viral protein interactions with cytokines – an example would be nice.

Response:

Thanks for your suggestion. Examples of "the viral protein interactions with cytokines" are in the Discussion section. (L376-378)

L306: Comparative RNA sequencing Analysis between of PK15 cells infected with PCV2 and cells treated with PCV2 + 5-Aza

Response:

Thank you for your correction. I've rewritten it. (L313)

L308: …shows that the expression of 3,340 genes was increased and the expression of 3,314 genes reduced…

Response:

Sorry for the confusing. I have corrected my mistake. (L315-316)

L311: “The KEGG enrichment analysis indicated that the varient genes in the drug treatment group and the PCV2 treatment group were primarily enriched in the cell cycle, the FoxO signalling pathway and the cytokine-cytokine receptor interaction” -> “The KEGG enrichment analysis indicates that various genes of the cell cycle, FoxO signalling pathway and cytokine cytokine receptor interaction are enriched in the drug treatment group and the PCV2 infection group.”

Response:

Thank you for reminding us. I've rewritten it. (L319-321)

L330: DEG – when used for the first time in the text write in full “differentially expressed genes”

Response:

Thanks for your advice. We have introduced the full name of “DEG” in its first use. (L191)

L348: despite of the availability

Response:

Thanks for your suggestion. We have replaced that paragraph. (L350-351)

L349 mutation in the PCV2 genome

Response:

Thank you for your correction. We have replaced that paragraph. (L351-352)

L350: PCV2 infection regulateds multiple of cellular processes including cytoskeleton turnover… - - - - - - - - - - - - - - - - - -

Response:

Sorry for the confusing. I have corrected the problem. (L376-377)

L 353 replication of in PK15 cells

Response:

Thank you for reminding us. I have corrected the problem. (L378-381)

L354: inflammation and apoptosis

Response:

Thanks for your advice. I have corrected the problem. (L380)

L355 and L400: “Mitogen-activated protein kinases (MAPKs) are necessary pathways…” – kinases are proteins and proteins cannot be pathways, please rephrase and don’t write the same sentence twice.

Response:

Thanks for your suggestion. The sentence in L400 has been deleted. I have revised the sentence. (L382)

L358: can activates 

Response:

Thank you for your correction. I have corrected the problem. (L384-385)

L361: …can reduce the downregulate biological processes such as transcription and translation of PCV2 genome into infected cells…

Response:

Sorry for the confusing. I have replaced the original sentence. (L388-389)

L367: the mRNA levels in of related cytokines…

Response:

Thank you for reminding us. I have replaced the original sentence. (L395-399)

L368: what does “functions related to inflammation and immunity” means?

Response:

Thanks for your advice. We have removed this sentence from the article. (L395-397)

L369: are also connected 

Response:

Thanks for your suggestion. I have replaced the original sentence. (L398)

L378: combined combining these this evidence….

Response:

Thank you for your correction. I have replaced the original sentence. (L404-415)

L380: the transcription and translation of the PCV2 genome

Response:

Sorry for the confusing. I have replaced the original sentence. (L389)

L381: …was carried out to assess…

L382: apoptosis and inflammation

L387: with the virus. and began

L390: the expressions of inflammatory and apoptosis factors were was detected

L391: the mRNA expressions levels of…

L392: apoptosis, factors

L395: leads to an obvious inflammatory response of in PK15 cells - -

L396: “After PCV2-infected PK15 cells added with 5-Aza, virus replication was accelerated, the cells produced more inflammation and apoptosis” -> “After addition of 5-Aza to PCV2 infected PK15 cells, virus replication as well as production of inflammatory cytokines by the cells and apoptosis were increased.”

L403: transcription and translation of PCV2 genome

Response:

Thank you for your valuable advice. Because of the redundant sentences of L381-403, I have deleted this paragraph. The above questions were included in the deleted paragraph.

L409: inflammation, and apoptosis

Response:

Thank you for reminding us. I have corrected the problem. (L420)

Comments on the Quality of English Language

English was mostly OK in the first part but some sentences in the discussion are not understandable (see word file). For the minor points/ English mistakes listed, note that I have probably missed some. Moreover, some things are formulated in an imprecise way which makes them wrong (mRNA expression instead of level, MAPKs are a pathway, translation of PCV2...). Therefore, please let a native speaker read and check the paper.

Response: 

Thanks very much for your valuable advice. I have rewritten the discussion section to focus on the issues you raised. Besides, we have this manuscript copyedited by a professional English editing service of MDPI Company that specializes in scientific papers, and the certificate is as follows:

Reviewer 4 Report (New Reviewer)

Comments and Suggestions for Authors

Yiyi Shan and colleagues present an interesting manuscript on the effects of 5-azacytidine on infection of kidney cells with porcine circovirus 2. The manuscript is generally well written, and the conclusions seem to be justified by the presented data. Nevertheless, there is not much information on the effects of 5-azacytidine documented so far for other viruses in the scientific literature, and the authors should include such information in the introduction and discussion parts.

 Minor remarks:

Line 40: please change to Circoviridae family

Lines 78-79: “Our study provides a new reference and method for the prevention and treatment of PCV2, and may provide...” The first part of the sentence can be misunderstood, please rephrase. I would suggest deleting the part “study provides a new reference and method for the prevention and treatment of PCV2, and “

Line 85: please change to “1 % penicillin streptomycin mixture”.

Lines 95-98: please rephrase.

line 123, 140: Please add the manufacturer.

Line 146: Regarding the fact that all steps of the western blotting procedure were described in detail: did you have a blocking step before incubating the membrane with the primary antibody?

Line 170: unclear, please rephase the part “and Annexin V/FITC was not added”.

Line 181: Could you please add some information on the primers or the kit used for cDNA synthesis?

Line 287: Do you mean 2397 and 1728 genes? Please rephrase.

Line 333: “Related genes were detected via qPCR”: please add information with the reference to the corresponding part of your results (I assume that you refer to 3.4.)

Comments on the Quality of English Language

The English language is generally well and understandable. Only minor corrections of the language might be necessary, especially in the discussion part.

Author Response

Reviewer 4 comments

Comments and Suggestions for Authors

Yiyi Shan and colleagues present an interesting manuscript on the effects of 5-azacytidine on infection of kidney cells with porcine circovirus 2. The manuscript is generally well written, and the conclusions seem to be justified by the presented data. Nevertheless, there is not much information on the effects of 5-azacytidine documented so far for other viruses in the scientific literature, and the authors should include such information in the introduction and discussion parts.

Sorry for the confusing. I have supplemented relevant references in the Introduction and Discussion sections to support my ideas. (L71-75 and L365-381)

Minor remarks:

Line 40: please change to Circoviridae family

Response:

Thanks for your advice. I have corrected my mistake. (L42)

Lines 78-79: “Our study provides a new reference and method for the prevention and treatment of PCV2, and may provide...” The first part of the sentence can be misunderstood, please rephrase. I would suggest deleting the part “study provides a new reference and method for the prevention and treatment of PCV2, and “

Response:

Thanks for your advice. I have replaced the original sentence. (L89-91)

Line 85: please change to “1 % penicillin streptomycin mixture”.

Response:

Sorry for the confusing. I've rewritten it. (L95)

Lines 95-98: please rephrase.

Response:

Thank you for reminding us. I have deleted the extra sentences. (L105-106)

line 123, 140: Please add the manufacturer.

Response:

Thank you for your valuable advice. I have added the information of the manufacturer. (L132 and L148-149)

Line 146: Regarding the fact that all steps of the western blotting procedure were described in detail: did you have a blocking step before incubating the membrane with the primary antibody?

Response:

Thanks for your advice. I've added the missing steps. (L157)

Line 170: unclear, please rephase the part “and Annexin V/FITC was not added”.

Response:

Sorry for the confusing. I've rewritten it. (L177-178)

Line 181: Could you please add some information on the primers or the kit used for cDNA synthesis?

Response:

Thank you for reminding us. I have added relevant information. (L132 and L139)

Line 287: Do you mean 2397 and 1728 genes? Please rephrase.

Response:

Thank you for your correction. I've rewritten it. (L295)

Line 333: “Related genes were detected via qPCR”: please add information with the reference to the corresponding part of your results (I assume that you refer to 3.4.)

Response:

Sorry for the confusing. I have corrected my mistake. (L339-340)

Comments on the Quality of English Language

The English language is generally well and understandable. Only minor corrections of the language might be necessary, especially in the discussion part.

Response: 

Thanks very much for your valuable advice. I have rewritten the discussion section to focus on the issues you raised.

Round 2

Reviewer 3 Report (New Reviewer)

Comments and Suggestions for Authors

Thank you for taking into account the comments, providing more information on the experiments, clarifying some unclear points like the virus titre in fig 2 and 3 and the subtype question. The discussion is also very much improved with suggestions as what to do in future work and open questions. And also the English is much better which makes it easier to follow and understand for the reader, thank you.

The only thing missing would be the positive control in figure 1 but maybe you may go without since it doesn't impact the rest of the paper and your conclusions and your findings are still very interesting to the field.

Reviewer 4 Report (New Reviewer)

Comments and Suggestions for Authors

Thank you for following the suggestions.

Comments on the Quality of English Language

There is only minor editing o fthe English language necessary, mainly typing errors.

This manuscript is a resubmission of an earlier submission. The following is a list of the peer review reports and author responses from that submission.

Round 1

Reviewer 1 Report

Comments and Suggestions for Authors

Lines 101-102: It would be better if the sentence could be revised into: "The DMEM complete medium was supplemented with 5-Aza at the final concentrations of 5 um/L, 10 um/L, 20 um/L, 30 um/L and 40 um/L".

Line 114: What was the concentrations of the primary and secondary antibodies used?

Line 123: It should be "cultured in 12-well plates" and “extracted with Novizan DNA extraction kit".

Lines 157-161: What was the concentrations of the primary and secondary antibodies used?

Line 183: It was mentioned that "flow cytometry showed the same results for intracellular ROS levels (Figure 1D)", but Figure 1D is not flow cytometry data, but photographs of the cells. This is very confusing to readers.

Figures 2C and 3C: Scale bars are missing.

Comments on the Quality of English Language

The manuscript needs more carful proofreading. There are grammatical errors or typos throughout the manuscript which can hinder the understanding of the scientific concepts.

Author Response

Response to reviewer’s comments:

  1. Lines 101-102: It would be better if the sentence could be revised into: "The DMEM complete medium was supplemented with 5-Aza at the final concentrations of 5 um/L, 10 um/L, 20 um/L, 30 um/L and 40 um/L".

Response:

Thanks for your advice. We have changed the sentence to: “The complete DMEM medium containing the final concentration of 5-Aza was re-placed with 5 μM, 10 μM, 20 μM, 30 μM and 40 μM,”. (Line 101-102)

  1. Line 114: What was the concentrations of the primary and secondary antibodies used?

Response:

Thank you for reminding us. we have supplemented the concentrations of the primary and secondary antibodies in the article: “The concentration of primary antibody was 2.66 μg/ml, and the concentration of secondary antibody was 125 μg/ml”. (Line 118-119)

  1. Line 123: It should be "cultured in 12-well plates" and “extracted with Novizan DNA extraction kit".

Response:

Thank you for your valuable advice. we have corrected language errors and company names as follows:

“Cells were cultured in 12-well plates, and extracted with Vazyme DNA extraction kit according to the instructions”. (Line 124)

  1. Lines 157-161: What was the concentrations of the primary and secondary antibodies used?

Response:

We have added: “The concentration of the primary antibody was 1 μg/ml, and the concentration of the secondary antibody was 0.02 μg/ml”. (Line 164-165)

  1. Line 183: It was mentioned that "flow cytometry showed the same results for intracellular ROS levels (Figure 1D)", but Figure 1D is not flow cytometry data, but photographs of the cells. This is very confusing to readers.

Response:

Sorry for not writing the experiment name clearly, which has been corrected as follows:

“Similarly, detection of cellular ROS with DCFH-DA probes showed the same results (Figure 1D)”. (Line 187-188)

  1. Figures 2C and 3C: Scale bars are missing.

Response:

Sorry for the confusing. In fact, the scale bars of Figures 2C and 3C are in the lower right corner of each figure. Due to the small scale, the original picture can be seen when enlarged.

Comments on the Quality of English Language

The manuscript needs more careful proofreading. There are grammatical errors or typos throughout the manuscript which can hinder the understanding of the scientific concepts.

Response:

Thank you for your advice. We have this manuscript copyedited by a professional English editing service of MDPI Company that specializes in scientific papers, and the certificate is as follows:

Thank you and all the reviewers for the kind advice. Should you have any questions, please contact us without hesitate. I prefer communication by email and I'm looking forward to hearing from you soon.

Sincerely yours,

Shenglong Wu

Key Laboratory for Animal Genetics, Breeding, Reproduction and Molecular Design of Jiangsu Province, College of Animal Science and Technology, Yangzhou University, Yangzhou, 225009 Jiangsu, China.

E- mail: [email protected]

Reviewer 2 Report

Comments and Suggestions for Authors

Porcine circovirus type 2 (PCV2) is a virus that targets pigs. It causes post-weaning multiple system failrue syndrome (PMWS), porcine dermatitis nephrotic syndrome, and porcine respiratory syndrome (PRDC), and porcine reproductive disorders. Preventing and treating PCV2 infection can save the pig industry a lot of money and improve meat supply. Authors here performed molecular and cellular assays to characterize the effect of 5-Aza treatment on PCV2 infection using a porcine kidney cell line as model.

Contrary to the expectation, 5-Aza treatment significantly increased the PCV2 viral copy number after infection compared to infection alone without the cytidine analogue. This is substantiated by both qPCR and immunoblot measurement of viral gene and protein expression level. Furthermore, authors found that the addition of 5-Aza further increased apoptotic gene expression during PCV2 infection, as well as a dramatic shift in host transcriptional pathways related to MAPK signaling pathway and other inflammatory response pathway.

Overall, this is a well-designed and interesting study, but multiple issues need to be addressed before publication. In particular, unit of measurement is not consistent throughout the manuscript, and the resolution of the figures is too low to evaluate the results. Number of replicates for each experiment should be specified. Additionally, the authors should consider including a treatment of only "5-Aza" with no PCV2 (mock) infection to evaluate the effect of "5-Aza" treatment using RNA-seq.

Specific suggestions for the manuscripts:

- line 61: "mi RNA" should be "microRNA (miRNA)"
- line 77: "Angle" should be "angle"
- line 177: "When the concentration of 5-Aza was below 10 um/L", is this a typo? Should it be "µg/mL"? Also present in lines 102, 179, 183, 187, 190, 191, 222, and other places in the manuscript. "um/L" should be "µg/mL". This could be a rendering issue with the document writer, so please double check the pdf before submission.
- Figure 1A: X-axis label says "um/L", should it be "µg/mL"?
- Figure 1: Please indiciate the type of statistical test and procedure performed in each figure caption.
- Figure 1B, 1C, 1D, 1E: Please indicate the concentration of 5-Aza used for each experiment in the figure caption.
- Figure 1F: Unclear what type of measurement was done to assess expression level. Is this qPCR? Indicate the number of replicates for each group, and how to derive the "Relative mRNA expression" from the raw measurement values?
- Line 204 - 205. The sentence starts "Table 15. cells infected ...". There are several words inserted here that don't seem to belong. Please check.
- Figure 2A: Please include a description for how viral copy number was established using qPCR in the methods. Include details about the standards used. This relates to Figure 2A as "PCV2-cap copies" is quantified, but the method for calibrating qPCR signal to copy number is not described.
- Figure 3: resolution of the figures is too low. The y-axis (especially the superscript in y-axis labels) is barely legible. Please increase the resolution of the figure.
- Figure 3A: Please indicate the number of replicates and independent experiments conducted to evaluate the effect.
- Figure 3B: A quantified value for the Cap protein expression should be provided. The Westernblot image is quite saturated and it is difficult to evaluate relative difference in the Cap expression between different treatments. Also "HSP90" should be shown on the bottom row as control, in line with the presentation of other Western blot images in this manuscript.
- Figure 4: Include a treatment of only "5-Aza" with no PCV2 (mock) infection to evaluate the effect of "5-Aza" treatment alone on apoptotic genes.
- Figure 4: Please indicate the number of replicates and independent experiments conducted to evaluate the effect.
- Figure 5, 6 and 7: Please increase resolution of the figures. Legends and axis labels are barely legible. Annotations are not legible.

Comments on the Quality of English Language

Please ensure consistency of units throughout the manuscript.

Author Response

Response to reviewer’s comments:

Porcine circovirus type 2 (PCV2) is a virus that targets pigs. It causes post-weaning multiple system failrue syndrome (PMWS), porcine dermatitis nephrotic syndrome, and porcine respiratory syndrome (PRDC), and porcine reproductive disorders. Preventing and treating PCV2 infection can save the pig industry a lot of money and improve meat supply. Authors here performed molecular and cellular assays to characterize the effect of 5-Aza treatment on PCV2 infection using a porcine kidney cell line as model.

Contrary to the expectation, 5-Aza treatment significantly increased the PCV2 viral copy number after infection compared to infection alone without the cytidine analogue. This is substantiated by both qPCR and immunoblot measurement of viral gene and protein expression level. Furthermore, authors found that the addition of 5-Aza further increased apoptotic gene expression during PCV2 infection, as well as a dramatic shift in host transcriptional pathways related to MAPK signaling pathway and other inflammatory response pathway.

Overall, this is a well-designed and interesting study, but multiple issues need to be addressed before publication. In particular, unit of measurement is not consistent throughout the manuscript, and the resolution of the figures is too low to evaluate the results. Number of replicates for each experiment should be specified. Additionally, the authors should consider including a treatment of only "5-Aza" with no PCV2 (mock) infection to evaluate the effect of "5-Aza" treatment using RNA-seq.

Response:

Thanks for your advice. 

We corrected the concentration units of the drugs, improved the resolution of the pictures, and indicated the number of replicates for each experiment.

In the preliminary experiment, we treated the cells with 10μM 5-Aza and determined cell viability, apoptosis, ROS and inflammatory cytokines, which confirmed that 5-AZA did not cause cell damage. Therefore, j2+5-Aza treatment group was not set up in subsequent experiments.

We will improve the experimental protocol and make it more rigorous in the future.

Specific suggestions for the manuscripts:

  1. line 61: "mi RNA" should be "microRNA (miRNA)"

Response:

Thank you for your correction. We have changed "mi RNA" to "microRNA (miRNA)". (Line 59)

  1. line 77: "Angle" should be "angle"

Response:

we have changed "Angle" to "angle". (Line 74)

  1. line 177: "When the concentration of 5-Aza was below 10 um/L", is this a typo? Should it be "µg/mL"? Also present in lines 102, 179, 183, 187, 190, 191, 222, and other places in the manuscript. "um/L" should be "µg/mL". This could be a rendering issue with the document writer, so please double check the pdf before submission.

Response:

Sorry for the confusing. We changed all concentration units in the article from “um/L” to “μM”.

  1. Figure 1A: X-axis label says "um/L", should it be "µg/mL"?

Response:

Thank you for reminding us. We have checked and modified the wrong unit name on the X-axis, changing "um/L" to "μM".

  1. Figure 1: Please indiciate the type of statistical test and procedure performed in each figure caption.

Response:

Thanks for your comment. In fact, we have indiciate the type of statistical test and procedures in the “2.9. Data Processing and Analysis” section of “Materials and methods”. As following:

The 2-ΔΔCt method was used to calculate the relative quantitative results, and the expression level was homogenized with internal reference genes. SPSS 25.0 software (SPSS, Inc., Chicago, IL, USA) was used for statistical analysis. The P-values were calculated using the two-group unpaired T-test or the one-way analysis of variance. All data are expressed as means ± SD for three repeated samples. After statistical treatment, P<0.05 indicated that the difference was statistically significant.

  1. Figure 1B, 1C, 1D, 1E: Please indicate the concentration of 5-Aza used for each experiment in the figure caption.

Response:

We have illustrated in 1B, 1C, 1D, 1E that the 5-Aza concentration used was 10μM. (Line 201-202)

  1. Figure 1F: Unclear what type of measurement was done to assess expression level. Is this qPCR? Indicate the number of replicates for each group, and how to derive the "Relative mRNA expression" from the raw measurement values?

Response:

Thanks for your comment. We have revised it.

In fact, we have indiciate the type of statistical test and procedures in the “2.9. Data Processing and Analysis” section of “Materials and methods”. As following:

The 2-ΔΔCt method was used to calculate the relative quantitative results, and the expression level was homogenized with internal reference genes. SPSS 25.0 software (SPSS, Inc., Chicago, IL, USA) was used for statistical analysis. The P-values were calculated using the two-group unpaired T-test or the one-way analysis of variance. All data are expressed as means ± SD for three repeated samples. After statistical treatment, P<0.05 indicated that the difference was statistically significant.

  1. Line 204 - 205. The sentence starts "Table 15. cells infected ...". There are several words inserted here that don't seem to belong. Please check.

Response:

Thank you for your correction. We have revised the sentence and removed “Table 15”: “In cells infected with PCV2, Cap expression was measured via qPCR, Western blot and IFA at 0, 12, 24, 36, 48 and 72 h”. (Line 210)

  1. Figure 2A: Please include a description for how viral copy number was established using qPCR in the methods. Include details about the standards used. This relates to Figure 2A as "PCV2-cap copies" is quantified, but the method for calibrating qPCR signal to copy number is not described.

Response:

Thanks for your comment. To investigate the effects of viral kinetics and drug treatment on viral replication at different time points, PCV2-Cap copy numbers were measured using relative quantification. Details has been described in the part of Data Processing and Analysis. For clarity, we changed the Y-axis name from "PCV2-cap copies" to "Relative levels of mRNA CAP expression".

  1. Figure 3: resolution of the figures is too low. The y-axis (especially the superscript in y-axis labels) is barely legible. Please increase the resolution of the figure.

Response:

Thank you for your advice. In fact, the resolution of the images in the original manuscript is 300 dpi, and now we have increased the resolution of some images to 1200 dpi.

  1. Figure 3A: Please indicate the number of replicates and independent experiments conducted to evaluate the effect.

Response: 

Thanks for your comment.

In fact, we have indiciate the type of statistical test and the number of replicates in the “2.9. Data Processing and Analysis” section of “Materials and methods”. As following:

The 2-ΔΔCt method was used to calculate the relative quantitative results, and the expression level was homogenized with internal reference genes. SPSS 25.0 software (SPSS, Inc., Chicago, IL, USA) was used for statistical analysis. The P-values were calculated using the two-group unpaired T-test or the one-way analysis of variance. All data are expressed as means ± SD for three repeated samples. After statistical treatment, P<0.05 indicated that the difference was statistically significant.

  1. Figure 3B: A quantified value for the Cap protein expression should be provided. The Westernblot image is quite saturated and it is difficult to evaluate relative difference in the Cap expression between different treatments. Also "HSP90" should be shown on the bottom row as control, in line with the presentation of other Western blot images in this manuscript.

Response: 

Thank you for reminding us. we have put the bands of “HSP90” below and we have marked the gray values below each sample in the picture, as shown: (Figure 3B)

  1. Figure 4: Include a treatment of only "5-Aza" with no PCV2 (mock) infection to evaluate the effect of "5-Aza" treatment alone on apoptotic genes.

Response:

Thank you for reminding us. In the preliminary experiment, we treated the cells with 10μM 5-Aza and determined cell viability, apoptosis, ROS and inflammatory cytokines, which confirmed that 5-AZA did not cause cell damage (Figure 1). Therefore, j2+5-Aza treatment group was not set up in subsequent experiments. We will improve the experimental protocol and make it more rigorous in the future.

  1. Figure 4: Please indicate the number of replicates and independent experiments conducted to evaluate the effect.

Response:

Thanks for your comment.

In fact, we have indiciate the type of statistical test and the number of replicates in the “2.9. Data Processing and Analysis” section of “Materials and methods”. As following:

The 2-ΔΔCt method was used to calculate the relative quantitative results, and the expression level was homogenized with internal reference genes. SPSS 25.0 software (SPSS, Inc., Chicago, IL, USA) was used for statistical analysis. The P-values were calculated using the two-group unpaired T-test or the one-way analysis of variance. All data are expressed as means ± SD for three repeated samples. After statistical treatment, P<0.05 indicated that the difference was statistically significant.

  1. Figure 5, 6 and 7: Please increase resolution of the figures. Legends and axis labels are barely legible. Annotations are not legible.

Response:

Thank you for your valuable advice. In fact, the resolution of the original picture in the manuscript is 300dpi, which may be blurred due to the compression of the picture in the document. We have now adjusted the Figures to 1200 dpi.

Comments on the Quality of English Language

Response:

Thank you for your advice. We have this manuscript copyedited by a professional English editing service of MDPI Company that specializes in scientific papers, and the certificate is as follows:

Please ensure consistency of units throughout the manuscript.

Response:

Sorry for the confusing. We changed all concentration units in the article from “um/L” to “μM”.

Thank you and all the reviewers for the kind advice. Should you have any questions, please contact us without hesitate. I prefer communication by email and I'm looking forward to hearing from you soon.

Sincerely yours,

Shenglong Wu

Key Laboratory for Animal Genetics, Breeding, Reproduction and Molecular Design of Jiangsu Province, College of Animal Science and Technology, Yangzhou University, Yangzhou, 225009 Jiangsu, China.

E- mail: [email protected]

Reviewer 3 Report

Comments and Suggestions for Authors

PCVD is a global porcine disease, which caused great ecnomic loss to the swine industry. Although vaccine was widely used to prevent and control this disease, higly genetic variation of this virus limit the use of vaccine. So, it is important to develop some drugs against PCV2 infection. 5-Azacytidine was used to evaluate its effect on the infection of PCV2 to PK-15 cells. They found the treatment of PK-15 cells with 5-AZ can increase the infection of this virus, and which associated with inflammation and MAPK signal. This study firstly revealed the roles played by 5-AZ in the infection of PCV2.

Some minor modifications were needed: 

1 Haemophilus should be changed Glaesserella, Mycoplasma should be italic, Angle should be angle or substitued with view.

2 The description for line 92-96 is inaccuracy and setence for line 98 is not complete.

3 The  unit for some u should be μ in the text.

4 The description for line 102 is inaccuracy, such as: at should be replaced with with, hole should be replaced with well.

5 The "made" in the line 108 should be replaced with "performed".

6 Some professional words were not rightly used, such as: membrane breaking, enclosed at room temperature, suggest a native speaker modification.

7 Novizan was wrong writed.

8 Deleting fefrigerator in the text.

9 Some cotent was repeated in line 158-159.

10 It is very aberrant in the description of line 204-205.

11 The flurorescen intensity of figure 2C is not clear and magnification is too low.

12 Discussion is a little superficial, which should closely associate with your results.

Comments on the Quality of English Language

English language should be further modified.

Author Response

Response to reviewer’s comments:

PCVD is a global porcine disease, which caused great ecnomic loss to the swine industry. Although vaccine was widely used to prevent and control this disease, higly genetic variation of this virus limit the use of vaccine. So, it is important to develop some drugs against PCV2 infection. 5-Azacytidine was used to evaluate its effect on the infection of PCV2 to PK-15 cells. They found the treatment of PK-15 cells with 5-AZ can increase the infection of this virus, and which associated with inflammation and MAPK signal. This study firstly revealed the roles played by 5-AZ in the infection of PCV2.

Some minor modifications were needed: 

1 Haemophilus should be changed Glaesserella, Mycoplasma should be italic, Angle should be angle or substitued with view.

Response:

Thanks for your comments. We have revised it (Line 48, 74).

2 The description for line 92-96 is inaccuracy and setence for line 98 is not complete.

Response:

Thank you for your valuable advice. We have revised the sentence according to your suggestion:

" Porcine kidney cells (PK15) (ATCC, CCL-33), in a 10% fetal bovine serum (FBS) (Gibco, Grand Island, NY, USA) and 1% streptomycin mixture (100U/ml penicillin, 0.1mg/ml streptomycin) (Beijing Solarbio Science & Technology Co., Ltd., Beijing, Chi-na) in DMEM medium (Gibco, Grand Island, NY, USA), were cultured in a 5% CO2 in-cubator at 37°C.".

“5-Aza (5-Azacytidine, ≥ 98% (HPLC)) (Aladdin, Shanghai, China) was dissolved in enzyme-free water”. (Line 105-112)

3 The unit for some u should be μ in the text.

Response:

Sorry for the confusing. We changed all concentration units in the article from “um/L” to “μM”.

4 The description for line 102 is inaccuracy, such as: at should be replaced with with, hole should be replaced with well.

Response:

Thank you for pointing out the deficiencies and we have revised the sentence words in question:

“The CCK-8 solution was added to the 96-well plate with 10 μL per well”. (Line 105)

5 The "made" in the line 108 should be replaced with "performed".

Response:

Thanks for your advice. We have amended it as follows:

“Optical density measurements were performed on a Tecan Infinit200 microplate reader (Sunrise, Tecan, Switzerland) at wavelength 450 nm”. (Line 107)

6 Some professional words were not rightly used, such as: membrane breaking, enclosed at room temperature, suggest a native speaker modification.

Response:

Thank you for your advice. We have this manuscript copyedited by a professional English editing service of MDPI Company that specializes in scientific papers, and the certificate is as follows:

7 Novizan was wrong writed.

Response:

We have changed the company name from "Novizan" to "Vazyme". (Line 124)

8 Deleting refrigerator in the text.

Response:

Thanks for your suggestion. I have deleted the word " in the fefrigerator" in the paper. (Line 127)

9 Some content was repeated in line 158-159.

Response:

Thank you for reminding us. We have removed the duplicate content. (Line 161)

10 It is very aberrant in the description of line 204-205.

Response:

Thank you for your comments, We have revised the sentence and removed “Table 15”. (Line 210)

11 The flurorescen intensity of figure 2C is not clear and magnification is too low.

Response:

Thank you for your advice. In fact, the resolution of the original picture in the manuscript is 1200 dpi, which may be blurred due to the compression of the picture in the document. (Figure 2)

12 Discussion is a little superficial, which should closely associate with your results.

Response:

Thanks for your advice. We have modified the discussion and conclusion section according to your suggestion to make it more relevant to my results.

Comments on the Quality of English Language

English language should be further modified.

Response:

Thank you for your advice. We have this manuscript copyedited by a professional English editing service of MDPI Company that specializes in scientific papers, and the certificate is as follows:

Thank you and all the reviewers for the kind advice. Should you have any questions, please contact us without hesitate. I prefer communication by email and I'm looking forward to hearing from you soon.

Sincerely yours,

Shenglong Wu

Key Laboratory for Animal Genetics, Breeding, Reproduction and Molecular Design of Jiangsu Province, College of Animal Science and Technology, Yangzhou University, Yangzhou, 225009 Jiangsu, China.

E- mail: [email protected]

Reviewer 4 Report

Comments and Suggestions for Authors

In this article you have explored :

"Effects and Mechanism Analysis of 5-Azacytidine on Porcine  
Circovirus Type 2 Infection"

however, it is not clear why this drug was chosen for exploration. Also, when you are describing the disadvantage of 5-Azacytidine, please do refer to some potential therapeutics which could be optional for PCV 2 infection treatments.

Author Response

(The authors gave the same response as above.)

Reviewer 5 Report

Comments and Suggestions for Authors

While the overall idea of studying the effect of 5-Aza on PCV2 replication is an interesting topic, the quality of English does not meet the minimum criteria and must be improved significantly.

Besides, there are several concerns which must be addressed:

1.      The study aimed to indicate mechanisms of the effects of 5-Aza on PCV2 infection, however, these mechanisms are not stated.

2.      The introduction section must be improved.

3.      What PCV2 strain was used? Must be specified.

4.      The time point after infection when PCV2-infected cells were collected is not specified. The molecular footprint of virus infection may significantly vary over the course of infection. Thus, even the PCV/mock data comparison may reveal only time-specific response.

5.      To evaluate the effect of 5-Aza on PCV2 infection, the control (non-infected PK-15) group must be treated with 5-Aza.

6.      The availability of PCV2 specific antibodies allows you to quantify virus replication. So, you could demonstrate the effect of the reagent on presence of live PCV2.

7.      There is an inconsistency in presenting PCV2 positive cells (green/red).

8.      Figure captions must be improved.

Comments on the Quality of English Language

 The quality of English does not meet the minimum criteria and must be improved significantly.

Author Response

Response to reviewer’s comments:

While the overall idea of studying the effect of 5-Aza on PCV2 replication is an interesting topic, the quality of English does not meet the minimum criteria and must be improved significantly.

Besides, there are several concerns which must be addressed:

  1. The study aimed to indicate mechanisms of the effects of 5-Aza on PCV2 infection, however, these mechanisms are not stated.

Response:

Sorry for the confusing. Through RNA sequencing analysis, we have identified the potential involvement of the MAPKs signaling pathway in regulating viral replication induced by 5-aza in PK15 cells, specifically through its impact on downstream target genes. However, a comprehensive exploration of the downstream key regulatory genes remains pending and will be addressed in future studies.

We have added the description of mechanisms in Conclusions section. (Line 422-424)

  1. The introduction section must be improved.

Response:

Thank you very much for giving us this opportunity to revise the paper. We have revised the introduction section according to your suggestion to make it more relevant to my results.

  1. What PCV2 strain was used? Must be specified.

Response:

Thanks for your advice. The virus subtype we used was PCV2d, which we have supplemented in the Materials and Methods section of the article.

  1. The time point after infection when PCV2-infected cells were collected is not specified. The molecular footprint of virus infection may significantly vary over the course of infection. Thus, even the PCV/mock data comparison may reveal only time-specific response.

Response:

Thank you for your suggestion. In Section 3.2 of this paper, we detected the PCV2 replication dynamic at different time points by qPCR, WB and IFA, and found the viral replication reached peak at 48h. Therefore, we chose 48h for subsequent experiments.

  1. To evaluate the effect of 5-Aza on PCV2 infection, the control (non-infected PK-15) group must be treated with 5-Aza.

Response:

Thank you for reminding us. In the preliminary experiment, we treated the cells with 10μM 5-Aza and determined cell viability, apoptosis, ROS and inflammatory cytokines, which confirmed that 5-Aza did not cause cell damage (Figure 1). Therefore, j2+5-Aza treatment group was not set up in subsequent experiments. We will improve the experimental protocol and make it more rigorous in the future.

  1. The availability of PCV2 specific antibodies allows you to quantify virus replication. So, you could demonstrate the effect of the reagent on presence of live PCV2.

Response:

Thank you for your valuable advice. We utilized PCV2-infected PK15 cells for drug treatment and did not evaluate the sole effect of 5-Aza on PCV2. However, it has been reported that "the primary antiviral mechanism of 5-Aza can be attributed to its capacity to enhance the frequency of HIV-1 mutations through viral DNA incorporation during reverse transcription [1]". 5-Aza possesses the ability to inhibit PCV2 genomic DNA methylation and augment the toxicity of PCV2.

In future investigations, we will further explore whether 5-Aza can integrate into the DNA of PCV2 and modify its toxicity.

[1]Dapp, M.;J.; Clouser, C.L.; Patterson, S.; Mansky, L.M. 5-Azacytidine can induce lethal mutagenesis in human immunodeficiency virus type 1. J Virol. 2009 Nov;83(22):11950-8. doi: 10.1128/JVI.01406-09.

There is an inconsistency in presenting PCV2 positive cells (green/red).

Response:

I apologize for any confusion caused. The IFA antibodies used in Figures 2 and 3 for PCV2-Cap were sourced from two different companies. Specifically, the primary antibody utilized in Figure 2C was obtained from VMRD, while the antibody employed in Figure 3C was acquired from abcam. It is possible that due to the variation in antibody sources, there are differences observed in the final images. Moving forward, we will place greater emphasis on this issue and ensure a more rigorous approach is taken during future research endeavors.

  1. Figure captions must be improved.

Response:

Thank you for your advice. We have revised the figure captions. 

Comments on the Quality of English Language

The quality of English does not meet the minimum criteria and must be improved significantly.

Response:

Thank you for your advice. We have this manuscript copyedited by a professional English editing service of MDPI Company that specializes in scientific papers, and the certificate is as follows:

Thank you and all the reviewers for the kind advice. Should you have any questions, please contact us without hesitate. I prefer communication by email and I'm looking forward to hearing from you soon.

Sincerely yours,

Shenglong Wu

Key Laboratory for Animal Genetics, Breeding, Reproduction and Molecular Design of Jiangsu Province, College of Animal Science and Technology, Yangzhou University, Yangzhou, 225009 Jiangsu, China.

E- mail: [email protected]

Round 2

Reviewer 2 Report

Comments and Suggestions for Authors

Thank you for the response to my comments and suggestions. I think the manuscript has been greatly improved as a result. There are still a few places that may require corrections. After addressing those issues, I think the manuscript should be published.

## Editing
- Line 44: "worldwidly" should be "worldwide"
- Figure 1F: please indicate the type of statistical test performed in figure caption (was it T-test?).
- Figures 2C and 3C: The scale bar is cropped out of the image. Please include the scale bar in the figure.
- Figure 3B: Figure sub-panel label "B" is cropped out
- Figure 3B: Are the values shown in this image correct? PCV2-Cap expression is "96.81" for sample treated with 10 µM of 5-Aza, but the Western blot band looks more intense than the other two treatments. Same issue with the values for HSP90. I suspect the expression values are shown in the wrong order. Also the values need to be shown with a unit. Please check.
- Line 247: extra dot on this line.

Comments on the Quality of English Language

Just a few minor edits required.

Author Response

Reviewer 2 comments

- Line 44: "worldwidly" should be "worldwide"

Response:

Thank you for your correction. We have changed " worldwidly " to " worldwide ". (Line 44)

- Figure 1F: please indicate the type of statistical test performed in figure caption (was it T-test?).

Response:

Thanks for your comment. We have indicated the type of statistical test performed in the Figure 1F.

- Figures 2C and 3C: The scale bar is cropped out of the image. Please include the scale bar in the figure.

Response:

Thank you for your advice. we've bolded the scale in the bottom right corner.

- Figure 3B: Figure sub-panel label "B" is cropped out

Response:

Thank you for your correction. We have revised the Figure sub-panel label "B"

- Figure 3B: Are the values shown in this image correct? PCV2-Cap expression is "96.81" for sample treated with 10 µM of 5-Aza, but the Western blot band looks more intense than the other two treatments. Same issue with the values for HSP90. I suspect the expression values are shown in the wrong order. Also the values need to be shown with a unit. Please check.

Response:

Thank you for your advice. The previous calculation was wrong, and we have corrected the gray value.

Reviewer 5 Report

Comments and Suggestions for Authors

Unfortunately, the major comments have not been addressed. Among them, the lack of transcriptomic data on PK-15 treated with 5-Aza limits the ability to make a clear conclusion. The fact that 5-Aza at certain concentration does not cause cell damage does not mean that 5-Aza is not able to change cellular metabolism.

Author Response

Reviewer 5 comments

Unfortunately, the major comments have not been addressed. Among them, the lack of transcriptomic data on PK-15 treated with 5-Aza limits the ability to make a clear conclusion. The fact that 5-Aza at certain concentration does not cause cell damage does not mean that 5-Aza is not able to change cellular metabolism.

Response: Thanks for your advice. Sorry for the inadequacies in the experimental design. In our study, to explore the effect of PCV2 on PK15 cells, we design 2 experimental groups, including normal PK15 cells (NC), PCV2 infected PK15 cells (PCV2). Considering the effect of 5-Aza on PK15 cells, we also added an experimental group, PCV2-infected + 5-Aza treatment (PCV2_5-Aza). Through the comparison between PCV2 group and PCV2_5-Aza group, we think it can also explain the effect of 5-Aza on PK15 cells to a certain extent. However, your advice is also very valuable. Due to the time limit of revision, we cannot complete the transcriptome sequencing in PK-15 treated with 5-Aza. Thus, we supplemented the expression verification of some DEGs (Figure 7C, as follows) in 5-Aza treated PK-15 cells. In the future design of the experiment, we also hope to continue to get your guidance.

Round 3

Reviewer 5 Report

Comments and Suggestions for Authors

In your later reply “the effect of PCV2 on PK15 cells” is mentioned as

1)      “Through the comparison between PCV2 group and PCV2_5-Aza group, we think it can also explain the effect of 5-Aza on PK15 cells to a certain extent”. Unfortunately, it is not sufficient to properly evaluate mechanisms underlying the effect 5-Aza on PCV2 infection. The major reason is that 5-Aza was used before PCV2 infection meaning that cells treated vs non-treated with 5-Aza were at different status before infection. Thus. The inappropriate experimental design limits the ability to address the major goal of this study.

2)      The study aimed to indicate mechanisms of the effects of 5-Aza on PCV2 infection, however, these mechanisms are not stated. Molecular footprint and mechanisms – are different concepts.  

3)      The introduction section does not fully explain why methylation is important for PCV2 replication.

4)      The title: “Effects and Mechanism Analysis of 5-Azacytidine on Porcine 1Circovirus Type 2 Infection” looks suboptimal.

5)      Simple summary – corrections of the Englich must be done.

6)      Line 43 – membraneless – non-enveloped.

7)      PCV2 must be spelled out when is used for the first time.

8)      Line 47: Simple PCV2 infection – what is that?

9)      Line 50: Moreover

10)  Studies have shown that PCV2 requires the help of genomic DNA in the nucleus to replicate

11)  Some sentences are difficult to understand. For example: Line 62: Studies have shown that PCV2 requires the help of genomic DNA in the nucleus to replicate.

12)  Line 59: Aberrant DNA methylation is often associated with tumorigenesis. – No reference provided. What is the relevance to your study?

13)  Line 63. How 5-Aza can be added to a genome? Genome of what?

14)  Line 66: Does your study involve using several PCV2 genotypes?

15)  Line 72: research – study.

16)  Overall, more work on the English is needed.

17)  Similarly, in M&M, line 107: “After addition of the primary antibody” – how they were added? Concentration? Incubation?

18)  M&M: please clarify the design experiment.

19)  Line 212: No need in using “virus” after “PCV2”

20)  Remains unclear what PCV2 strain was used.

21)  What PCV2 strain was used? Must be specified.

22)  The time point after infection when PCV2-infected cells were collected is not specified. The molecular footprint of virus infection may significantly vary over the course of infection. Thus, even the PCV/mock data comparison may reveal only time-specific response.

Comments on the Quality of English Language

see above

Author Response

Reviewer 5 comments

1) “Through the comparison between PCV2 group and PCV2_5-Aza group, we think it can also explain the effect of 5-Aza on PK15 cells to a certain extent”. Unfortunately, it is not sufficient to properly evaluate mechanisms underlying the effect 5-Aza on PCV2 infection. The major reason is that 5-Aza was used before PCV2 infection meaning that cells treated vs non-treated with 5-Aza were at different status before infection. Thus. The inappropriate experimental design limits the ability to address the major goal of this study.

Response: Thanks very much for your valuable advice. To explore mechanisms underlying the effect of 5-Aza on PCV2 infection, we analyzed the transcriptome difference among NC group, PCV2 group and PCV2_5-Aza group. In addition, there are similar groups of studies (Xu et al., 2023), such as NC group, DON group, melatonin (MEL)+DON group.

Xu Y, Xie Y, Wu Z, Wang H, Chen Z, Wang J, Bao W. Protective effects of melatonin on deoxynivalenol-induced oxidative stress and autophagy in IPEC-J2 cells. Food Chem Toxicol. 2023 Jul;177:113803.

2) The study aimed to indicate mechanisms of the effects of 5-Aza on PCV2 infection, however, these mechanisms are not stated. Molecular footprint and mechanisms – are different concepts.  

Response: Thanks for your advice. In “Simple summary” and “Simple summary” section, we have added the mechanism description of the effects of 5-Aza on PCV2 infection.

3) The introduction section does not fully explain why methylation is important for PCV2 replication.

Response: Thanks for your advice. We have revised the introduction section.

4) The title: “Effects and Mechanism Analysis of 5-Azacytidine on Porcine Circovirus Type 2 Infection” looks suboptimal.

Response: Thanks for your advice. We have revised the title: “Identification of Effect of 5-Azacytidine on Porcine Circovirus Type 2 Replication in Porcine Kidney Cells”.

5) Simple summary – corrections of the Englich must be done.

Response: Thanks for your advice. We have revised the “Simple summary” section.

Besides, we have this manuscript copyedited by a professional English editing service of MDPI Company that specializes in scientific papers, and the certificate is as follows:

6) Line 43 – membraneless – non-enveloped.

Response: Thank you for your correction. We have changed “membraneless” to “non-enveloped”.

7) PCV2 must be spelled out when is used for the first time.

Response: Thank you for your suggestion. When we first used “PCV2”, we added an explanation for “PCV2”. (Line 21)

8) Line 47: Simple PCV2 infection – what is that?

Response: Sorry for the confusing. We have changed “simple” to “single”.

9) Line 50: Moreover

Response: Thanks for your advice. We removed “moreover”.

10) Studies have shown that PCV2 requires the help of genomic DNA in the nucleus to replicate. 11)Some sentences are difficult to understand. For example: Line 62: Studies have shown that PCV2 requires the help of genomic DNA in the nucleus to replicate.

Response: Sorry for the confusing. We have changed the sentence to: “Studies have shown that the genome of PCV2 is single-stranded circular DNA, and its complementary strand is usually encoded in host cells [11]”.

12) Line 59: Aberrant DNA methylation is often associated with tumorigenesis. – No reference provided. What is the relevance to your study?

Response: Thank you for your correction. We have removed this sentence.

13) Line 63. How 5-Aza can be added to a genome? Genome of what?

Response: We have revised this sentence as follows: “As a classical DNA methylation inhibitor, 5-Aza can bind to DNA methyltransferase in the receptor genome and inhibit DNA methylation [12,13]”.

14) Line 66: Does your study involve using several PCV2 genotypes?

Response: Thanks for your advice. In this study, porcine kidney cells were used as a model to study the impacts of 5-Aza in PCV2d infection. And we have changed the sentence you are confused about.

15) Line 72: research – study.

Response: Thank you for your correction. We have changed “research” to “study”.

16) Overall, more work on the English is needed.

Response: Thanks for your advice. In the process of revision, we have this manuscript copyedited by a professional English editing service of MDPI Company that specializes in scientific papers, and the certificate is as follows:

17) Similarly, in M&M, line 107: “After addition of the primary antibody” – how they were added? Concentration? Incubation?

Response: Sorry for the confusing. We have modified the content of this part: “bovine serum albumin (BSA) (Solarbio, Beijing, China) was added and blocked for 2h at 37℃. BSA was discarded, the primary antibody (PCV2 Cap antibody, VMRD, Pull-man, WA, USA) was added, and the cells were incubated in a refrigerator at 4°C over-night. The concentration of primary antibody was 2.66 μg/ml, and the concentration of sec-ondary antibody was 125 μg/ml”.

18) M&M: please clarify the design experiment.

Response: Thanks for your advice. In M&M section, we have added some description. Such as “2.9. Reactive Oxygen Species Assessment”, “2.10. Flow Cytometry Analysis”, “2.11. RNA-Sequencing”.

19) Line 212: No need in using “virus” after “PCV2”

Response: Sorry for the confusing. We have revised it in the full paper.

20)Remains unclear what PCV2 strain was used.

Response: Sorry for the confusing. We have added the PCV2 strain: PCV2d.

21) What PCV2 strain was used? Must be specified.

Response: Sorry for the confusing. We have added the PCV2 strain: PCV2d.

22) The time point after infection when PCV2-infected cells were collected is not specified. The molecular footprint of virus infection may significantly vary over the course of infection. Thus, even the PCV/mock data comparison may reveal only time-specific response.

Response: Thanks for your advice. In Figure 2, we detected the PCV2 replication dynamic at different time points (0, 12, 24, 36, 48 and 72 h) measured via qPCR, WB and IFA. Results showed, the level of viral replication reached peak at 48h. Thus, we selected PK15 cells with PCV2 infection for 48 h, which was used for transcriptome sequencing. However, your advice is valuable. Considering the time limit of revision, we will design the time point in the future research.
